# Evaluation of Thin Film Composite Forward Osmosis Membranes: Effect of Polyamide Preparation Conditions

Aya Mohammed Kadhom [1], Mustafa Hussein Al-Furaiji [2], Zaidun Naji Abudi [1]

[1] Environmental Engineering Department, College of Engineering, Mustansiriyah University, Iraq
[2] Environment and Water Directorate, Ministry of Science and Technology, Baghdad, Iraq

*Correspondence to*: Mustafa Al-Furaiji (alfuraiji79@gmail.com)

**Abstract** The forward osmosis (FO) process has been considered for desalination as a competitor option to the traditional reverse osmosis process. Interfacial polymerization (IP) reaction between two monomers (i.e., m-phenylene diamine (MPD) and 1, 3, 5-benzenetricarbonyl chloride (TMC)) is typically used to prepare the selective polyamide layer that prevents salts and allows water molecules to pass. In this research, we investigated the effect of preparation conditions (MPD contact time, TMC reaction time, and addition of an amine salt) on the FO performance in terms of water flux and salt flux. The results showed that increasing MPD contact time resulted in a significant increase in the water flux and salt flux. However, increasing TMC reaction time caused a decline in both the water flux and the salt flux. The optimum condition that gave the highest water flux (64 L.m$^{-2}$.h$^{-1}$) was found to be as 5 min for MPD and 1 min for TMC. The addition of an amine salt of camphorsulfonic acid-triethylamine (CSA-TEA) was able to make an apparent effect on the FO process by increasing water flux (74.5 L.m$^{-2}$.h$^{-1}$).

**Keywords:** Forward Osmosis; Thin-Film Composite; Polysulfone; Interfacial Polymerization; Polyamide

## 1. Introduction

Water Purification is the process of removing pollutants from raw water to produce water for human consumption (drinking water) or other beneficial purposes (irrigation, livestock, and industrial use) (Maddodi et al., 2020). Membrane processes are among the most effective methods that can be used for water purification especially for desalination of water.

At this time, the most effective technique is the reverse osmosis (RO) process, where it can be used to desalinate seawater and also for wastewater reuse (Kadhom et al., 2019; Kalash et al., 2020). RO can be defined as the process that relies on external force, in which the applied hydraulic pressure is responsible for transporting water through the membrane (Peñate and García-Rodríguez, 2012).

Forward osmosis (FO) is an osmotically driven membrane process that uses the osmotic pressure gradient to drive water transport across a semi-permeable membrane while rejecting most solutes (Cath et al., 2006; McCutcheon et al., 2005). In the FO process, water transports from a low osmotic pressure solution (i.e., feed solution) to a higher osmotic pressure solution (i.e., draw solution). Besides, FO has been considered a high water recovery and low-cost purification option compared to the pressure-driven membrane processes like reverse osmosis (Linares et al., 2017). One of the most critical factors affecting the development of the FO process is preparing a suitable membrane for the process. The ideal membranes for FO have to be able to provide high water permeability, high rejection of solutes, substantially reducing internal concentration polarization (ICP), and has high chemical stability and mechanical strength (Ren and McCutcheon, 2014; Zhao et al., 2012).

Thin-film composite (TFC) membranes have been studied widely for FO applications (Al-Furaiji et al., 2019; Chowdhury et al., 2017; Ren and McCutcheon, 2017). TFC membranes consist of two layers: a selective layer that only allows water to pass and rejects salt and a support layer that gives the membrane the required mechanical properties. Most of the FO studies

on TFC membranes have been focusing on developing the support layer, while fewer studies have been considering improving the selective layer.

The preparation of the polyamide selective layer is conducted using interfacial polymerization (IP) reaction (Mohammadifakhr et al., 2020). Typically, the IP reaction occurs between two reactive monomers: m-phenylene diamine (MPD) in the aqueous phase with 1, 3, 5-benzenetricarbonyl chloride (TMC) in the organic phase (Raaijmakers and Benes, 2016). Previous studies have reported that controlling the IP reaction conditions could significantly affect the performance of the formed polyamide layer (Kadhom and Deng, 2019a) however, most of these studies were dealing with reverse osmosis membranes (Dong et al., 2015; Jin and Su, 2009; Zhao et al., 2013). In contrast, very few studies have investigated the effect of interfacial polymerization reaction on the performance of the TFC FO membranes (Klaysom et al., 2013). Therefore, studying the effect of the preparation conditions can help in preparing highly efficient FO membranes.

In this work, the effect of m-phenylenediamine (MPD) aqueous solution exposure time and trimesoyl chloride (TMC) organic solution reaction time is studied. Besides, the effect of incorporating an amine salt to the MPD solution was reported. This paper aims to study the conditions of the interfacial polymerization reaction on the efficiency of the TFC membranes in the FO process. Scanning electron microscopy (SEM), atomic force microscopy (AFM), and contact angles measurements were used to characterize the prepared membranes.

## 2. Materials and Methods

### 2.1. Materials

Polysulfone (PSU, MW= 22000) from Xian Lyphar Biotech, China, was used to fabricate membranes substrates. N,N dimethylformamide (DMF, 99.8%) and 2,2,4-trimethylpentane (isooctane, 99%) were purchased from Fluka Chemie AG,Buchs, Switzerland. M-phenylenediamine (MPD, >99%) and trimesoyl chloride (TMC, 98%) were ordered from Merck. Triethylamine (TEA, ≥99%), and (1s)- -10-(+)camphorsulfonic acid (CSA, 99%), were purchased from Sigma Aldrich. Sodium chloride (NaCl) was purchased from Thomas Baker, India. Deionized water (DI water) was used to prepare NaCl and MPD aqueous solutions and for other purposes such as cleaning.

### 2.2. Preparation of PSU support layer

The phase inversion method was used to prepare PSU supporting sheets. The casting solution was prepared by dissolving 17 wt. % dry polysulfone pellets in DMF. The mixture was stirred and heated to 60°C for 6 h until a clear solution was formed, which was then degassed for more than 24 h at room temperature before use. Afterward, the solution was cast using a home-made casting knife by taking an aliquot from the clear solution by a pipette to spread on a clean glass plate to the desired thickness. The glass plate with the solution was then immersed into a water bath at room temperature resulting in the immediate formation of the PSU support sheet that was separated from the glass plate in a moment. Then, all of the sheets were collected and stored in DI water for 24 h or more at 4°C before use.

### 2.3. Preparation of TFC membrane

TFC Forward osmosis membranes were fabricated on the top surface of the PSU sheet by interfacial polymerization reaction between MPD aqueous solution and TMC organic solution. MPD aqueous solution was prepared by dissolving 2% MPD in DI water while the TMC solution was made by dissolving 0.15% of TMC in isooctane. Firstly the MPD solution was poured onto the PSU sheet at different contact times. Then, the TMC solution was poured onto the PSU sheet that is containing the MPD active sites and the reaction time was also varied to study the effect of IP reaction time. At first, the MPD contact time was varied from 2 to 5 min with keeping the contact time of the TMC at 1 min. Then, the best MPD contact time (i.e. 5 minutes) that gave the highest water flux was chosen and the TMC solution contact time was studied in the range of 1 to 4

minutes. In order to study the effect of adding CSA-TEA at a weight ratio of 2:1, they were added to the MPD solution with a weight percent of 1%. The IP reaction was conducted at room temperature. Finally, the obtained TFC membranes were dried in the oven at 60°C for 10 min and then collected and stored in DI water for 24 h until testing.

### 2.4. Membrane characterization

Scanning Electron Microscope (Fesem Tescan Mira3 France) and Atomic Force Microscope (Angstrom advanced Inc., 2008, U.S.A) were used to determine the morphology of the prepared membrane. The hydrophilicity of the membranes was measured using contact angles (Theta Lite TL-101 Thailand).

### 2.5. FO performance test

The FO performance was tested in a bench-scale system, as shown in Figure 1. This system consists of two tanks: one of them is used for the feed solution and the other contains the draw solution. DI water was used as a feed solution while 1 M NaCl was used as a draw solution based on the standard methodology that was described by (Cath et al., 2013). These solutions were pumped to the membrane cell using two pumps from Pure-water (model: 75GPD, volts: 24VDC, workflow: 28LPH). All experiments were conducted in FO mode (i.e., active layer faces the feed solution). The membrane was installed in a custom-made cell with the chamber's dimensions of length 7.62 cm, width 2.54 cm and a depth of 0.3 cm. Water flux $J_w$ can be estimated using the following equation (Al-Furaiji et al., 2018):

$$J_w = \frac{\Delta V}{At}$$

Where $J_w$ is the water flux (LMH: $Lm^{-2} h^{-1}$), $\Delta V$ is the change in feed solution volume (L), A is the active area of the membrane ($m^2$), and t is the experiment's time (h).

Typically, salt flux is used in forward osmosis investigations to describe the selectivity of the membrane, while salt rejection is normally used in reverse osmosis studies. The salt rejection equation can be used when there is a feed solution involved in the process, while in FO, there are feed solution and draw solution. That is why the salt flux is used instead of the salt rejection. Salt flux through the membrane was estimated by monitoring the change in conductivity of the feed solution and using the following equation (Al-Furaiji et al., 2020):

$$J_s = \frac{\Delta CV}{At}$$

Where $J_s$ is the salt flux (GMH: $gm^{-2} h^{-1}$), $\Delta C$ is the change in the feed solution concentration (g/L), V is the volume of feedwater flow (L), A is the active area of the membrane ($m^2$), and t is the experiment's time (h).

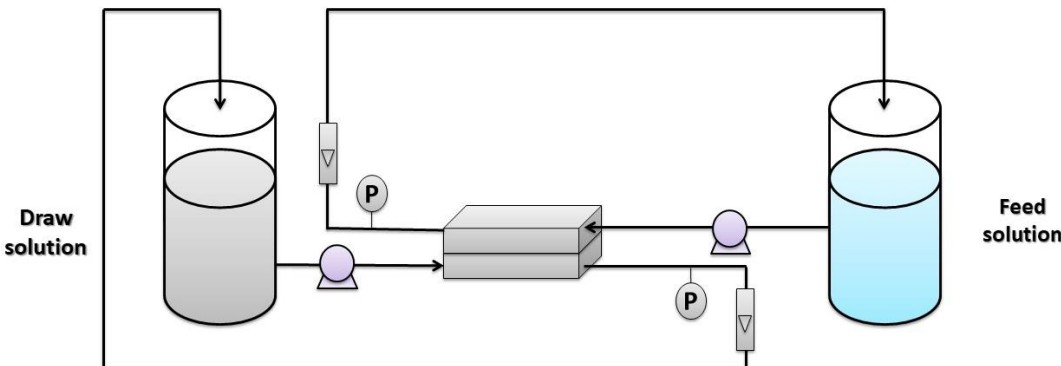

106                **Figure 1. Schematic diagram of the FO bench-scale test unit. (Al-Furaiji et al., 2018)**

107

## 3. Results and discussion

### 3.1. Membrane characterization

Two dimensional (2D) and three dimensional (3D) AFM images of the top surface of the TFC membrane are presented in a scan area of 2500 x 2500 nm, as shown in Figure 2. It can be clearly seen that the surface of the polyamide layer has a ridge-and-valley structure with an average roughness of the FO membrane surface is 5.07 nm. The surface roughness of the prepared membrane was quite similar to what has been reported for typical FO (Mi and Elimelech, 2008) and Nanofiltration membranes (Li et al., 2017, 2020). The rougher surface could be more beneficial for membrane performance as it gives higher areas for mass transfer especially when dealing with low fouling feed solution.

SEM images were used to investigate the surface morphology of the TFC membranes, as shown in Figure 3. It can be seen that the polyamide selective membrane was successfully formed on the PSU support sheet. This was confirmed by the leaf-like morphology which is a typical structure for polyamide TFC membranes (Kadhom and Deng, 2019b; Zhou et al., 2014).

Figure. 4 shows the contact angle measurement of the PSU support membrane and the polyamide selective layer. The contact angle of the PSU sheet was about 65°, while the thin polyamide layer had a contact angle of 33°. Contact angle of the membrane can be influenced by many parameters such as monomer concentration, reaction time, type of organic solution, post-treatment condition, etc. during IP reaction process. However, the reported value of the contact angle in this research lies within the range of the previously reported contact angle of TFC membrane (Kadhom et al., 2016; Lau et al., 2015). When the contact angle is small, that means the membrane is hydrophilic, meaning that the water penetrates easily into the pores of the membrane, so that gives a better osmotic water flux performance. Nevertheless, when the contact angle is large, it means that the film is hydrophobic, so the internal surfaces of the pores get dry. That gives a membrane with lower water flux.

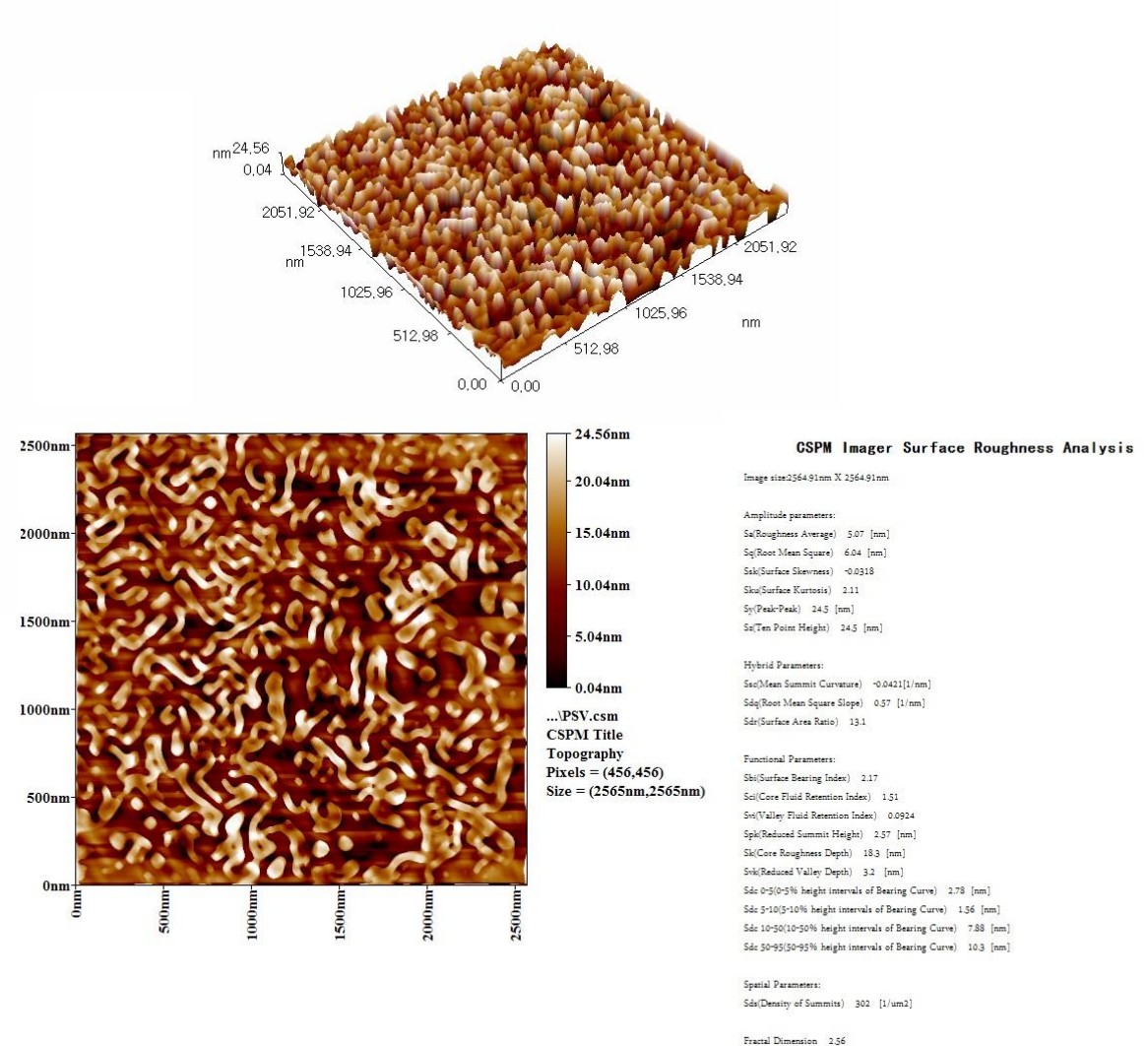


Figure 2. AFM images of the TFC-FO membrane.


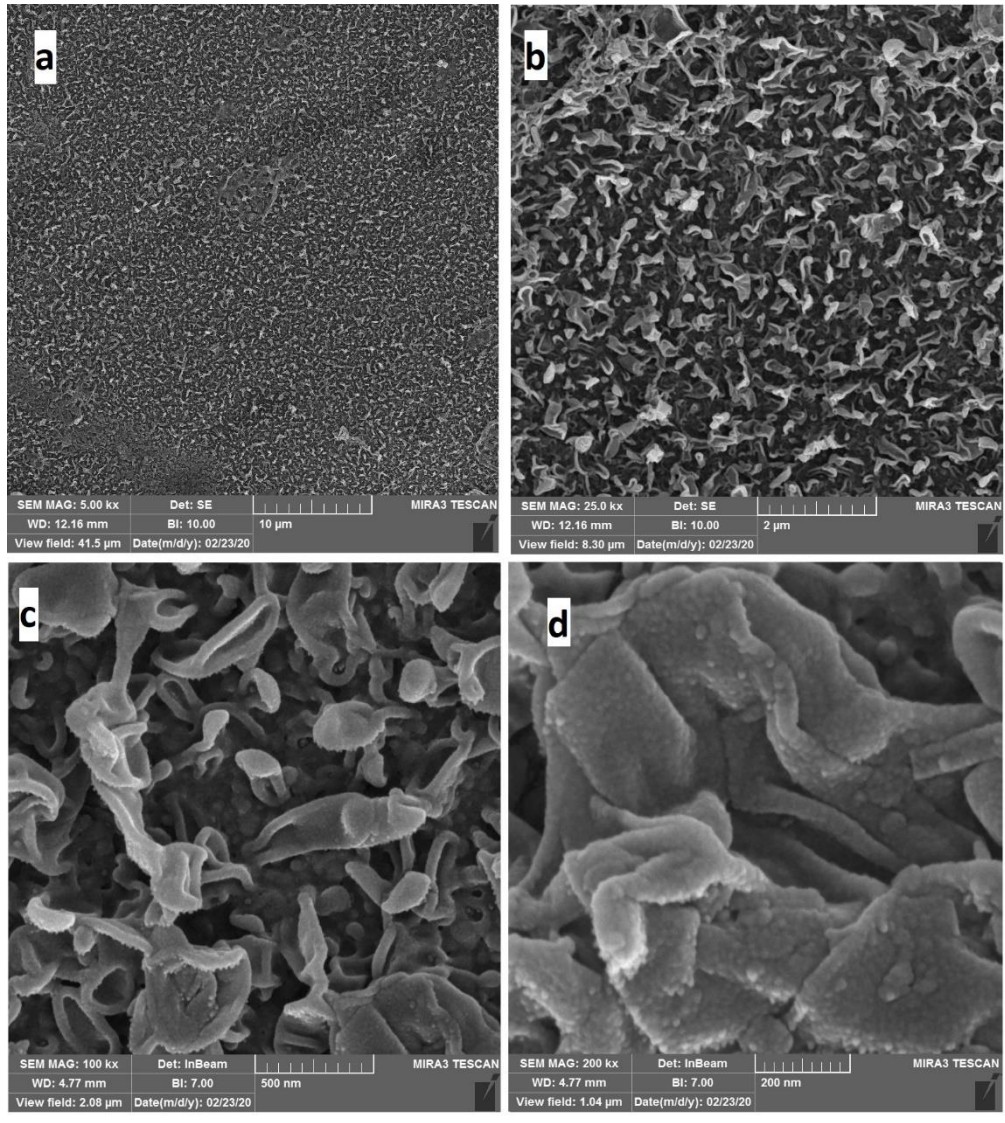


**Figure 3. SEM images of the TFC-FO membrane: a) 5000X, b) 25000X, c) 100000X, and d) 200000X.**

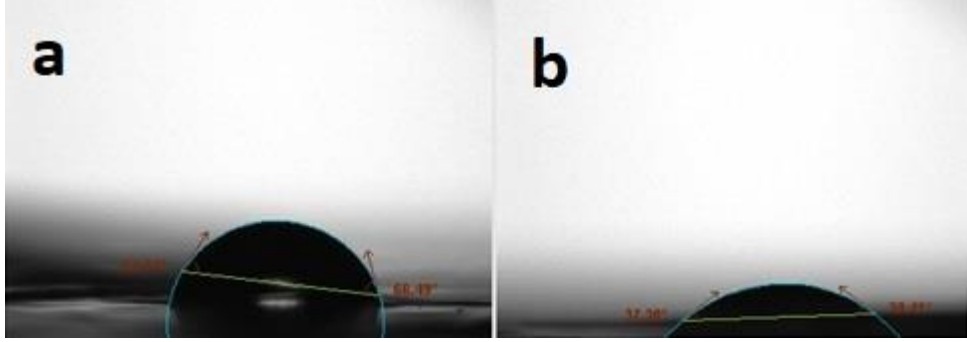


**Figure 4. The contact angle of a) PSU substrate membrane and b) polyamide thin layer.**

### 3.2. Effect of MPD exposure time

After the PSU support sheet has been prepared, the active layer is prepared by pouring the MPD solution onto the PSU layer
after fixing it well on a glass plate. The effect of MPD exposure time on the performance of the TFC-FO membrane was
studied by varying the contact time from 2 to 5 minutes while fixing the TMC reaction time at 1 minute, as shown in Figure
140 5.

The osmotic performance results revealed that water flux increases when the MPD exposure time increases. Also, increasing the MPD contact time leads to increasing salt flux. In fact, higher MPD exposure time means more MPD molecules would react with the support layer and accordingly increasing the IP reaction active sites. Besides, well-formed crosslinking would be achieved at higher MPD exposure time, which gives better IP reaction conditions when reacts with the TMC later (Kadhom and Deng, 2019a).

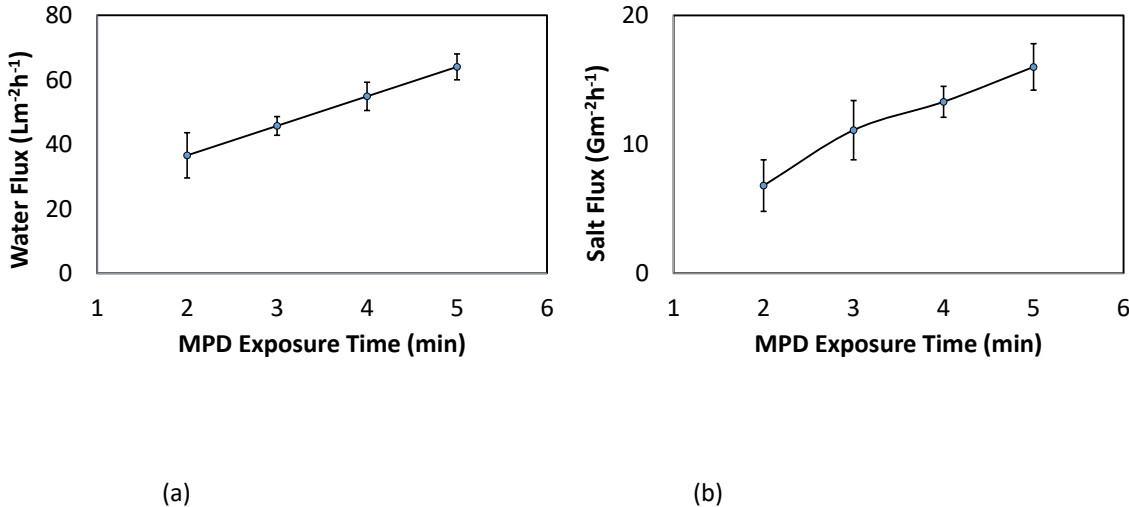

(a)                                                            (b)

**Figure 5. The effect of MPD contact time on membrane performance.** Feed solution: DI water and draw solution: 1M NaCl. **(a) The water flux (LMH) changing with different MPD exposure time (min.). (b) The salt flux (GMH) changing with different MPD exposure time (min.).**

### 3.3. Effect of TMC reaction time

In order to study the effect of IP reaction time, TMC contact time was varied from 1 to 4 min with fixing the MPD exposure time at 5 min, as shown in Figure 6. TMC organic solution was poured on the PSU substrate that contains the MPD active sites to conduct the IP reaction. It can be seen that the optimum condition that gave the highest water flux was recorded at a reaction time of 1 min. Interestingly, it can also be noticed that water flux and salt flux decreased sharply with increasing the TMC reaction time. This is mainly attributed to that increasing the TMC contact time leads to generating a thicker polyamide layer and consequently higher mass transfer resistance to permeation of water(Zhou et al., 2014). Moreover, the extent of the cross-linking is increased with increasing the IP reaction time and as a result, water flux and salt flux decreased (Wang et al., 2017).

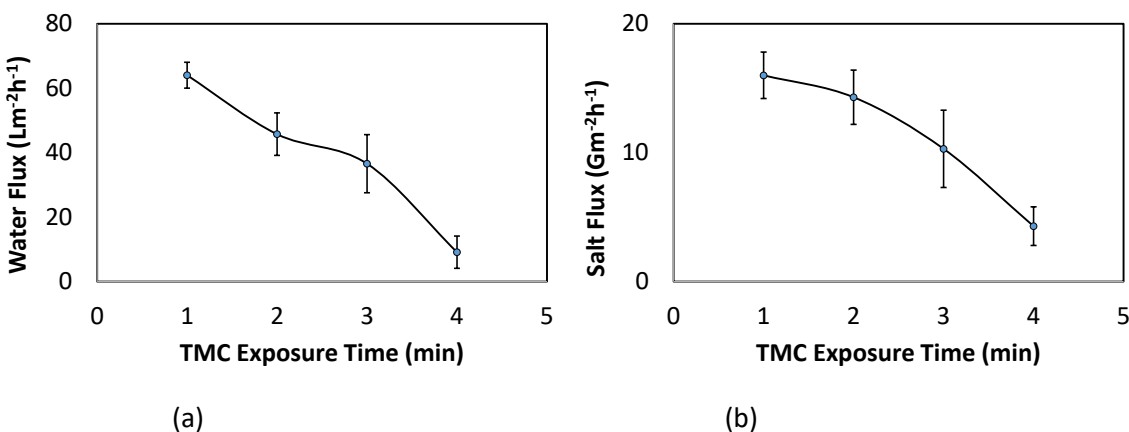

(a)                                                            (b)

**Figure 6. The effect of TMC contact time on membrane performance.** Feed solution: DI water and draw solution: 1M NaCl. **(a) The water flux (LMH) changing with different TMC exposure time (min.). (b) The salt flux (GMH) changing with different TMC exposure time (min.).**

### 3.4. Effect of CSA-TEA salt

The effect of adding an amine salt (i.e., CSA-TEA) on the performance of the FO process was studied as shown in Figure 7. It has been found that adding 1% of the CSA-TEA to the aqueous MPD solution exhibited a moderate increase in both water flux and salt flux. Similar behavior was reported for reverse osmosis and Nanofiltration processes (Khorshidi et al., 2017). It is known that polyamide formation during the IP reaction can result in the release of hydrogen chloride (Raaijmakers and Benes, 2016). The formation of hydrogen chloride can affect the reactivity of the monomer reactant in the aqueous phase (i.e., MPD). Therefore, the addition of a strong base such as TEA enhances the reactivity of the MPD and consumes the produced acid (i.e. hydrogen chloride). Also, it has been reported that the TEA acts as a catalyst by accelerating the MPD–TMC reaction and generating thinner and more crosslinked polyamide layer (Vatanpour et al., 2017; Wang et al., 2017).

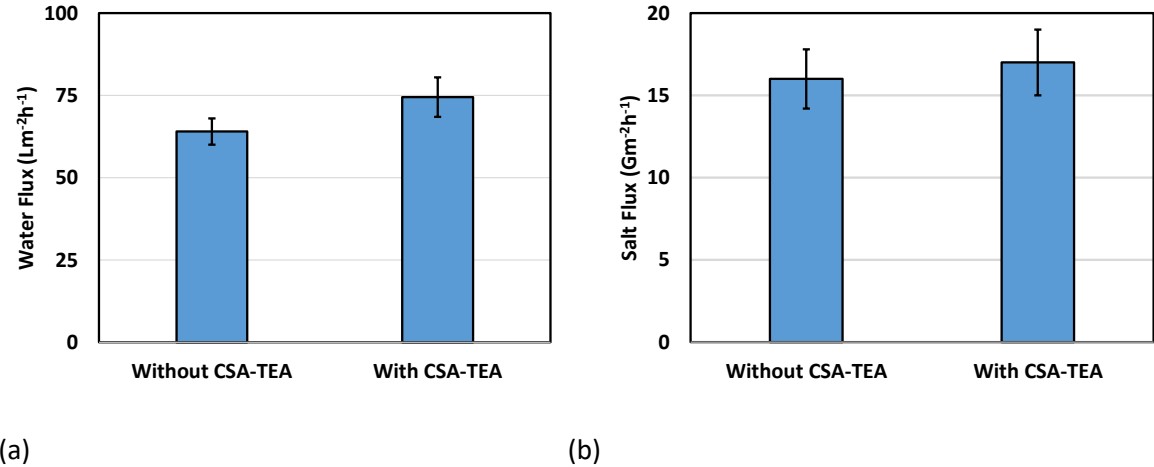

(a)                                                              (b)

**Figure 7. The effect of adding CSA-TEA to the MPD-aqueous solution on a) water flux and b) salt flux.** Feed solution: DI water and draw solution: 1M NaCl.

A comparison of the TFC-PSU membrane with some of the previously reported TFC membranes can be found in Table1. It can be seen that TFC-PSU membrane exhibited the highest water flux compared to the reported membranes. However, the reverse salt flux value lies within the range of the previously reported salt flux of the TFC membranes. If we look closely to the results of our previous work (Al-Furaiji et al., 2020) and compare it to the current work, it can be distinguished that the water flux of the current work is about twice that of the previous work, while the salt flux is a bit higher. There are two main differences between the previous work (Al-Furaiji et al., 2020) and the current work:

1. In the previous work, we used PAN polymer as a support for the TFC FO membrane, while in this work, we used PSU polymer.
2. In the previous work, the support layer was prepared using the electrospinning method while in this work phase inversion method was used.
The polyamide layer was perfectly formed and well distributed on the PSU support layer compared to the PAN nanofibers based membrane. This is most likely due to the smaller pore size and the hydrophobic nature of the PSU substrate. Although, electrospinning method produces a highly porous membrane, but phase inversion makes a more robust membrane that can perform better in FO testing.

**Table 1. Comparison of the performance of some TFC membranes from previous studies.**

| Membrane | Feed solution | Draw solution | Water flux (L/m² h) | Salt flux (G/m² h) | Reference |
|---|---|---|---|---|---|
| TFC-PSU | DI water | 1 M NaCl | 36.58 | 6.8 | This work. |
| HTI-TFC | DI water | 1 M NaCl | 15 | 4.5 | (Ren and McCutcheon, 2014) |
| TFC-PAN | DI water | 1 M NaCl | 16 | 4 | (Al-Furaiji et al., 2020) |
| Aquaporin TFC | DI water | 1 M NaCl | 9 | 4 | (Xia et al., 2017) |

| | | | | | |
|---|---|---|---|---|---|
| TFC-M2 (CAB substrate) | DI water | 1 M NaCl | 16.8 | 5.88 | (Ma et al., 2020) |
| TFC-CTA (HTI, commercial) | DI water | 1 M NaCl | 12.0 | 8.04 | (Kwon et al., 2017) |
| CAB | DI water | 1 M NaCl | 9.0 | 3.78 | (Ong et al., 2012) |
| PVDF nanofiber-PA | DI water | 1 M NaCl | 11.6 | 3.48 | (Tian et al., 2013) |
| PSU /Silica-PA | DI water | 1M NaCl | 31 | 7.44 | (Liu and Ng, 2015) |
| Oasys TFC | DI water | 1M NaCl | 30 | 50 | (Cath et al., 2013) |

## 4. Conclusion

In this work, TFC forward osmosis membranes were prepared on PSU substrate (17wt %) as a support layer via IP reaction between MPD and TMC to form a polyamide selective layer. The effect of MPD and TMC reaction times was investigated. The best results were found to be at 5 min for MPD and 1 min for TMC reaction times. These results gave the best performance of FO membranes in terms of water flux and salt rejection. Increasing MPD exposure time leads to increasing the active sites on the PSU layer. By changing the TMC reaction time, it is possible to control how dense the polyamide layer and, consequently, the amount of water and salt that passes through the membrane. Also, the effect of adding an amine salt (CSA-TEA) on the performance of FO membranes was demonstrated. The result showed that moderate improvement in water flux was achieved. Finally, this study can be considered as a useful guide for researchers and workers in the field of preparing TFC forward osmosis. Future research can focus on investigating other additives to the MPD and TMC solutions. Also, studying the effect of changing MPD and TMC concentrations in preparing TFC-FO is highly recommended for future works.

## Declaration of competing interest

The authors declare that they have no known competing financial interests or personal relationships that could have appeared to influence the work reported in this paper.

## Acknowledgements

The author would like to thank Mustansiriyah University, Baghdad, Iraq, and the Ministry of Science and Technology in Iraq for their support in the present work.

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
