# Peer review of "Evaluation of Thin Film Composite Forward Osmosis Membranes"

_Drinking Water Engineering and Science, 2020_

## Referee Comment (RC1) · Anonymous Referee #1 · 31 Oct 2020

Comments I have reviewed the manuscript entitled "Evaluation of Thin Film Composite Forward Osmosis Membranes: Effect of Polyamide Preparation Conditions". I recommend minor revision; though, the following comments need to be addressed. 1- The language is generally good; though, I recommend another round of revision. 2- Abs., Please identify the performance results at the optimum conditions. 3- CSA-TEA (2:1), is this a weight or mole percentage? 4- Figure 1, if you used this figure from another work, please cite. 5- Line 116, "while the thin polyamide layer had a contact angle of 33o." Please explain why this contact angle is lower than the similarly prepared TFC membrane. 6- Is it possible to draw the salt rejection with the salt flux? 7- What is the percentage of the salt in figure 7? Please add. 8- Please make a table to compare this

work's results with similar works.

---

## Referee Comment (RC2) · Anonymous Referee #2 · 9 Nov 2020

The authors presented the effect of exposure time of MPT and TMC on the water/salt flux in the prepared FO membranes. From the desalination point of view, an optimal FO membrane should have high water flux but low salt flux. Why did the authors concluded that the best results were found to be at 5 min for MPD and 1 min for TMC reaction times (highest water and salt fluxes)? In the figures, please avoid using abbreviations like LMH, GMH. In the authors publication: M. Al-Furaiji et al.: TFC membranes supported with nanofibers for forward osmosis process, the water and salt flux reported is much lower as compared with the values presented in this manuscript. What drives such differences? If we zoom-in to compare the water flux and salt flux reported in M. Al-Furaiji et al.: TFC membranes supported with nanofibers for forward osmosis

process (previous work) and in current work, the water flux is approx 4 time higher than that reported in previous work, but the salt flux is approx 6-8 time higher than that reported in previous work. This means that the salt rejection by the FO membrane prepared in the current work will be significantly lower than the membrane prepared in your previous work. it will be interesting comparison to be discussed in the manuscript.

---

## Author Comment (AC1) · 9 Nov 2020

**Dear Reviewer,**

The authors graciously acknowledge the reviewer's comments on our manuscript. We provide responses to each comment received below. Our response is given in red.

Comments I have reviewed the manuscript entitled "Evaluation of Thin Film Composite Forward Osmosis Membranes: Effect of Polyamide Preparation Conditions". I recommend minor revision; though, the following comments need to be addressed.

1- The language is generally good; though, I recommend another round of revision.

We have gone through the manuscript thoroughly again to English-improve the text by re-writing some parts and correcting grammatical errors and typos. We believe that the text in general has improved in this new version.

2- Abs., Please identify the performance results at the optimum conditions.

The abstract has been modified to address reviewer's suggestion.

3- CSA-TEA (2:1), is this a weight or mole percentage?

This is weight ratio; this has been clarified in the manuscript.

4- Figure 1, if you used this figure from another work, please cite.

A reference was added to Figure 1.

5- Line 116, "while the thin polyamide layer had a contact angle of 33o." Please explain why this contact angle is lower than the similarly prepared TFC membrane.

[revised manuscript text omitted]

---

## Author Comment (AC2) · 20 Nov 2020

Dear Reviewer, We appreciate your valuable comments on our manuscript and the fruitful discussion points that you have raised; below are our answers to your comments: The authors presented the effect of exposure time of MPT and TMC on the water/salt flux in the prepared FO membranes. • From the desalination point of view, an optimal FO membrane should have high water flux but low salt flux. Why did the authors concluded that the best results were found to be at 5 min for MPD and 1 min for TMC reaction times (highest water and salt fluxes)? Even though the salt flux increased when water flux increased (at 5 min for MPD and 1 min for TMC), but the

salt flux still within the acceptable limit where the Js/Jw ratio is 0.25 g/L compared to what has been reported in the literature. So, we concluded that this membrane was the optimum as it provided the highest water flux with a salt flux of an acceptable value. • In the figures, please avoid using abbreviations like LMH, GMH. The figures will be updated in the next version, according to the reviewer's comment. • In the authors publication: M. Al-Furaiji et al.: TFC membranes supported with nanofibers for forward osmosis process, the water and salt flux reported is much lower as compared with the values presented in this manuscript. What drives such differences? If we zoom-in to compare the water flux and salt flux reported in M. Al-Furaiji et al.: TFC membranes supported with nanofibers for forward osmosis process (previous work) and in current work, the water flux is approx 4 time higher than that reported in previous work, but the salt flux is approx 6-8 time higher than that reported in previous work. This means that the salt rejection by the FO membrane prepared in the current work will be significantly lower than the membrane prepared in your previous work. it will be interesting comparison to be discussed in the manuscript. We appreciate the reviewer's comments. To compare our previous TFC membrane with the current one, we should compare both membranes at the same preparation conditions (MPD= 2min, and TMC= 1min.); please see the following table. Water flux Salt flux This work 35.58 $\pm$7 6.8 $\pm$2 Previous work 16 $\pm$1.5 4 $\pm$0.5

It can be seen that the water flux of the current work is about twice that of the previous work, while the salt flux is a bit higher. There are two main differences between the previous work and the current work: 1. In the previous work, we used PAN polymer as a support for the TFC FO membrane, while in this work, we used PSU polymer. 2. In the previous work, the support layer was prepared using the electrospinning method while in this work phase inversion method was used. The polyamide layer was perfectly formed and well distributed on the PSU support layer compared to the PAN nanofibers based membrane. This is most likely due to the smaller pore size and the hydrophobic nature of the PSU substrate. Although, electrospinning method produces a highly porous membrane, but phase inversion makes a more robust membrane that

can perform better in FO testing.

Please also note the supplement to this comment:
https://dwes.copernicus.org/preprints/dwes-2020-33/dwes-2020-33-AC2-supplement.pdf

---

## Referee Comment (RC3) · Anonymous Referee #1 · 21 Nov 2020

Thanks to the authors for sufficiently answering my comments.

At this point, I have no other comments and recommend publishing the manuscript.

---

## Author Response (AR2)

**Comments to the Author:**

Thanks for revising the manuscript. However, not all comments were addressed in the revised manuscript (such as comment 5 and 6 of reviewer 1). In addition, there is no reference to the "previous work" in the added text.

Dear Editor,

We appreciate your comments on our manuscript. The answers to comment 5 and 6 of reviewer 1 were added to the text. Also, the reference to the "previous work" was added to the text and to the reference list.

**Dear Reviewer 1,**

The authors graciously acknowledge the reviewer's comments on our manuscript. We provide responses to each comment received below. Our response is given in red.

I have reviewed the manuscript entitled "Evaluation of Thin Film Composite Forward Osmosis Membranes: Effect of Polyamide Preparation Conditions". I recommend minor revision; though, the following comments need to be addressed.

1- The language is generally good; though, I recommend another round of revision.

We have gone through the manuscript thoroughly again to English-improve the text by re-writing some parts and correcting grammatical errors and typos. We believe that the text in general has improved in this new version.

2- Abs., Please identify the performance results at the optimum conditions.

The abstract has been modified to address reviewer's suggestion.

3- CSA-TEA (2:1), is this a weight or mole percentage?

This is a weight ratio; this has been clarified in the manuscript.

4- Figure 1, if you used this figure from another work, please cite.

A reference was added to Figure 1.

5- Line 116, "while the thin polyamide layer had a contact angle of 33o." Please explain why this contact angle is lower than the similarly prepared TFC membrane.

[revised manuscript text omitted]

**Dear Reviewer 2,**

We appreciate your valuable comments on our manuscript and the fruitful discussion points that you have raised; below are our answers to your comments. Our response is given in red.:

The authors presented the effect of exposure time of MPT and TMC on the water/salt flux in the prepared FO membranes.

- From the desalination point of view, an optimal FO membrane should have high water flux but low salt flux. Why did the authors concluded that the best results were found to be at 5 min for MPD and 1 min for TMC reaction times (highest water and salt fluxes)?

Even though the salt flux increased when water flux increased (at 5 min for MPD and 1 min for TMC), but the salt flux still within the acceptable limit where the $J_s/J_w$ ratio is 0.25 g/L compared to what has been reported in the literature. So, we concluded that this membrane was the optimum as it provided the highest water flux with a salt flux of an acceptable value.

- In the figures, please avoid using abbreviations like LMH, GMH.

The figures will be updated in the next version, according to the reviewer's comment.

- In the authors publication: M. Al-Furaiji et al.: TFC membranes supported with nanofibers for forward osmosis process, the water and salt flux reported is much lower as compared with the values presented in this manuscript. What drives such differences? If we zoom-in to compare the water flux and salt flux reported in M. Al-Furaiji et al.: TFC membranes supported with nanofibers for forward osmosis process (previous work) and in current work, the water flux is approx 4 time higher than that reported in previous work, but the salt flux is approx 6-8 time higher than that reported in previous work. This means that the salt rejection by the FO membrane prepared in the current work will be significantly lower than the membrane prepared in your previous work. it will be interesting comparison to be discussed in the manuscript.

We appreciate the reviewer's comments. To compare our previous TFC membrane with the current one, we should compare both membranes at the same preparation conditions (MPD= 2min, and TMC= 1min.); please see the following table.

|  | Water flux | Salt flux |
|---|---|---|
| This work | 35.58 ±7 | 6.8 ±2 |
| Previous work | 16 ±1.5 | 4 ±0.5 |

It can be seen that the water flux of the current work is about twice that of the previous work, while the salt flux is a bit higher. There are two main differences between the previous work and the current work:

1. In the previous work, we used PAN polymer as a support for the TFC FO membrane, while in this work, we used PSU polymer.

2. In the previous work, the support layer was prepared using the electrospinning method while in this work phase inversion method was used.

The polyamide layer was perfectly formed and well distributed on the PSU support layer compared to the PAN nanofibers based membrane. This is most likely due to the smaller pore size and the hydrophobic nature of the PSU substrate. Although, electrospinning method produces a highly porous membrane, but phase inversion makes a more robust membrane that can perform better in FO testing.